# Structure of a seeded palladium nanoparticle and its dynamics during the hydride phase transformation

Ana F. Suzana [1✉], Longlong Wu[1], Tadesse A. Assefa[1,6], Benjamin P. Williams [2], Ross Harder[3], Wonsuk Cha [3], Chun-Hong Kuo[4], Chia-Kuang Tsung [2,7] & Ian K. Robinson[1,5✉]

Palladium absorbs large volumetric quantities of hydrogen at room temperature and ambient pressure, making the palladium hydride system a promising candidate for hydrogen storage. Here, we use Bragg coherent diffraction imaging to map the strain associated with defects in three dimensions before and during the hydride phase transformation of an individual octahedral palladium nanoparticle, synthesized using a seed-mediated approach. The displacement distribution imaging unveils the location of the seed nanoparticle in the final nanocrystal. By comparing our experimental results with a finite-element model, we verify that the seed nanoparticle causes a characteristic displacement distribution of the larger nanocrystal. During the hydrogen exposure, the hydride phase is predominantly formed on one tip of the octahedra, where there is a high number of lower coordinated Pd atoms. Our experimental and theoretical results provide an unambiguous method for future structure optimization of seed-mediated nanoparticle growth and in the design of palladium-based hydrogen storage systems.

[1] Condensed Matter Physics and Materials Science Department, Brookhaven National Laboratory, Upton, NY, USA. [2] Department of Chemistry, Merkert Chemistry Center, Boston College, Chestnut Hill, MA, USA. [3] Advanced Photon Source, Argonne National Laboratory, Lemont, IL, USA. [4] Institute of Chemistry, Academia Sinica, Taipei, Taiwan. [5] London Centre for Nanotechnology, University College London, London, UK. [6] Present address: Stanford Institute for Materials and Energy Sciences, SLAC National Accelerator Laboratory, Menlo Park, California, USA. [7] Deceased: Chia-Kuang Tsung.
✉email: asuzana@bnl.gov; irobinson@bnl.gov

Hydrogen has the highest chemical energy per unit mass compared with other chemical fuels, three times larger than hydrocarbons fuels[1], and it is considered a relevant technology, having a great potential to limit environmental impacts. How to safely store hydrogen remains a key goal for the advancement of this new technology. One promising hydrogen storage strategy is the use of certain metals, which can form hydrides and reversibly absorb large amounts of this gas. Hydrogen can be absorbed by palladium (Pd), and this reaction is characterized by three steps: hydrogen molecular dissociation followed by chemisorption of hydrogen atoms on the Pd surface; next, hydrogen diffusion into the Pd subsurface and, lastly, diffusion into the bulk metallic structure[2]. The Pd-hydrogen system ($PdH_x$) can be described by the formation of two distinct phases, whose existence depends on the hydrogen concentration: the hydrogen-poor solid-state solution called α phase, and the hydrogen-rich hydride, the so-called β phase. At room temperature (RT), the maximum hydrogen solubility in the α phase is reached when $x = 0.017$, while the single β phase exists for $x > 0.6$, and intermediate $x$ values represent the coexistence of both phases[3]. The formation of the β phase is accompanied by a lattice expansion of ~3.5% in the Pd crystalline structure, caused by the absorption of a high number of hydrogen atoms, forming a new crystalline structure. This α–β phase transformation in Pd nanoparticles is of special scientific interest because it shows fast kinetics under ambient conditions, and is a model system for optimization of the efficiency of hydrogen storage materials.

It is well known that nanostructured materials present significant thermodynamic deviations from their bulk counterparts, which is mainly attributed to their larger surface area to volume ratio[4]. It is generally unclear if the observed deviations come directly from the diminutive size effect or from inhomogeneities induced in the sample by its surfaces. The majority of studies on the interstitial dissolution of hydrogen in metals leading to phase transformations have been done on ensemble sample formats in the 14–110 nm size range[5], prone to heterogeneity, which results in averaged data, disregarding the influence of the size, shape and, crystallinity on the phase transformation process. Further insight can be obtained by measuring single particles under in situ conditions to identify the role of the nanostructure on the chemical reaction. The phase transformation of the $PdH_x$ system has been extensively studied for bulk samples, but the deviations for nanosized individual nanoparticles are only now starting to be investigated, partly due to the recent development of appropriate techniques, such as electron microscopy for characterization of 40 nm[6], 80 nm[7], and 20–74 nm[8] nanoparticles, and plasmonic spectroscopy for 30 nm[9] and 17–100 nm nanoparticles[10]. However, none of these techniques can characterize both the morphology and 3D displacement distribution at the same time during the α–β phase transformation. Here, we report a coherent diffraction experiment that shows this transformation, with picometer sensitivity of the lattice displacements.

A common and efficient approach used for the synthesis of nanoparticles is the so-called seed-mediated growth, which is based on two steps: preparation of the seed and the subsequent injection of this solution as a precursor for the growth of the larger nanoparticles. This method is based on the reduction of the ionic form of the metal precursor onto the seed surface through the use of a reducing agent and it routinely permits the control of both the size and the shape of the final metal nanoparticles. Capping agents are frequently adopted as they can selectively adsorb onto a specific crystal facet of the seed particles and consequently slow down the growth of that facet. The resulting shape of the nanocrystals can be controlled by the nature of the seed and the synthesis conditions, such as reducing/capping agents' concentrations, temperature, etc. However, the research into the detailed mechanism of how the seed

is incorporated into the final nanoparticles is still lacking. For example, the induced distortion or strain distribution of the final nanoparticle containing the seed has not been measured yet, to the best of our knowledge.

Bragg coherent diffraction imaging (BCDI) is a lensless technique that works by illuminating a sample with a coherent X-ray probe and recording the coherent diffraction pattern of a single Bragg peak from an individual nanocrystal[11,12]. For nanosized objects, the coherent diffraction patterns contain fringes due to the finite size of the object, but in a diffraction experiment, only the amplitude of the complex Fourier is measured, not its phase. The missing phase of the diffracted radiation is recovered iteratively using phase retrieval algorithms[13], going back and forth between the reciprocal and real spaces, until convergence is reached between the measured data and the reconstruction. In this process, constraints in each space are adopted: the real space constraint includes a support region where the electron density of the object is allowed to exist, and the reciprocal space constraint consists of the solution having the observed amplitude of the oversampled diffraction pattern[14]. The amplitude of the resulting 3D complex image is proportional to the electron density of the crystal contributing to the Bragg peak and the phase is the projection of the displacement field onto the Bragg vector of the crystalline structure compared with the ideal position of their atoms[15]. The technique can reach a nanometric real-space resolution, usually 10–20 nm[16], while the distortions caused by strain are given with subangstrom sensitivity[15]. The possibility of using reactive environments during the measurement makes BCDI an ideal technique to probe dynamic changes and observe the evolution of defects' distribution. In recent years, in situ BCDI has revealed the dynamics occurring during catalytic processes[17–19], strain mapping and nucleation dislocation during ionic diffusion in Lithium-ion batteries[20–24], as well as the defect evolution during the hydriding phase transformation of palladium nanocrystals[25–27]. Ulvestad et al.[25] investigated the dynamics of cubic Pd nanoparticles during the α–β phase transformation using BDCI and the results are consistent with hydrogen uptake in the cube corners, followed by the nucleation of the β phase in one corner of the nanocube. Very similar dynamics were also observed for cubic Pd nanoparticles using electron microscopy[7,28].

In this work, we use the seed-mediated growth methodology for producing octahedral Pd nanocrystals, rather than the cubes studied by Ulvestad et al.[25]. We utilize BCDI for measuring the displacement distribution of an individual nanoparticle in three dimensions under pristine conditions at RT and show that the seed is not only visible but also plays a crucial role in distorting the final crystalline structure of the nanoparticle. This resulting distortion is consistent with finite element analysis (FEA) calculations. To understand the effect of the nanocrystal shape on the phase transition dynamics, the same nanocrystal is measured under a hydrogen partial pressure of 4410 Pa and the hydride phase transformation is directly visualized as a function of time. We captured the partial transformation into the β phase and show that the diffusivity of the hydrogen atoms into the Pd crystalline structure is coupled with the coordination number and, possibly, lattice displacement. Our results give structural insights into the reaction states as a function of time at the nanoparticle level, showing that the β phase is preferentially formed in one corner of the octahedron, following the "cap model" already predicted by the theoretical study shown by Ulvestad et al.[26].

## Results and discussion
### Imaging the displacement field of one single Pd particle under pristine conditions. Pd colloidal nanoparticles were synthesized

following a seed-mediated growth approach ("Methods"). In this process, firstly small cubic Pd seeds are synthesized and then used as a precursor for the growth of the larger octahedral nanoparticles through its injection into a $Pd^{2+}$ salt solution and further reduction of this metallic cation by the gradual addition of an ascorbic acid solution. In this case, cetyltrimethylammonium bromide (CTAB) molecules were used as the capping agent, leading to residual strain on the nanocrystal surface. A bright-field transmission electron microscopy (TEM) image of the Pd seed is shown in Supplementary Fig. 1a. One major challenge of BCDI measurements is that nanocrystals can move under the beam due to radiation pressure[29]. In order to address this issue, the Pd nanoparticles were coated with $TiO_2$ as fixed support to reduce their mobility under the beam. After synthesis, the nanoparticles in colloidal dispersion were drop-casted onto a flat substrate and coated with crystalline $TiO_2$ using atomic layer deposition (ALD), which permits precise control over the layer thickness of the surrounded material. The $TiO_2$ thickness was maintained around 100 nm, as shown in the scanning electron microscopy (SEM) image in Supplementary Fig. 1b. Faceted octahedral nanocrystals with sizes in the range of 200–300 nm are clearly seen in the SEM image of the as-prepared sample. The boundaries between the metallic nanoparticles and the oxide layer are visible, with the former showing brighter contrast. For the hydrogen chemisorption into the Pd surface to occur, the hydrogen molecules must diffuse through the $TiO_2$ layer. Our results below show little effect of steric hindrance provoked by the oxide layer. As the hydrogen atoms are small, their diffusion into the Pd crystalline structure is apparently not prevented by the presence of the oxide coating. A detailed description of the synthesis and ALD procedures is given in the Methods section.

Figure 1a shows an SEM image of an individual nanocrystal surrounded by the oxide coating. A schematic with the octahedral geometrical representation of the nanocrystal is shown as the black lines, separating the brighter/darker SEM contrast areas. Palladium nanoparticles with octahedral shape have been chosen due to their crystalline nature, where each crystal is confined by eight {111} planes, where four facets meet at each of the six {100} vertices of the octahedron. This system is used in this work as a prototype for the α–β phase transformation because it is expected to reveal the effect of facets and vertices on the transition dynamics. The BCDI experiment was performed at the 34-ID-C beamline at an advanced photon source (APS). In brief, the focused coherent X-rays illuminate the Pd nanoparticles inside a gas environmental cell that contains the nanoparticles. A photon-counting area detector recorded the diffraction patterns while the crystal angle was rocked through the Pd (111) Bragg peak.

Figure 1b shows the 3D Bragg peak for the nanoparticle measured in its pristine state ($p_{H_2} = 0$), under a pure nitrogen gas atmosphere, at RT. The sharp fringes demonstrate an adequate oversampling ratio in the diffraction pattern. Two different views of the reconstruction of the nanocrystal in the pristine state are displayed in Fig. 1c, visualized as an isosurface using the open-source application *Paraview*[30], and it is consistent with a single-crystalline octahedral shape, in accordance with the electron microscopy images. The angles between all the facets were estimated to be 71 ± 2°. The image phase values were sufficiently strong that unwrapping was required, using the code described in the "Methods", and the original reconstruction with wrapped phases is shown in Supplementary Fig. 2. The black arrow denotes the [111] direction of the momentum transfer vector, Q, which is perpendicular to one of the octahedral facets. The scale of the 3D images is calibrated to represent the relative local displacement field, projected along the Q-vector, where positive values (red-colored) are attributed to atomic displacements along

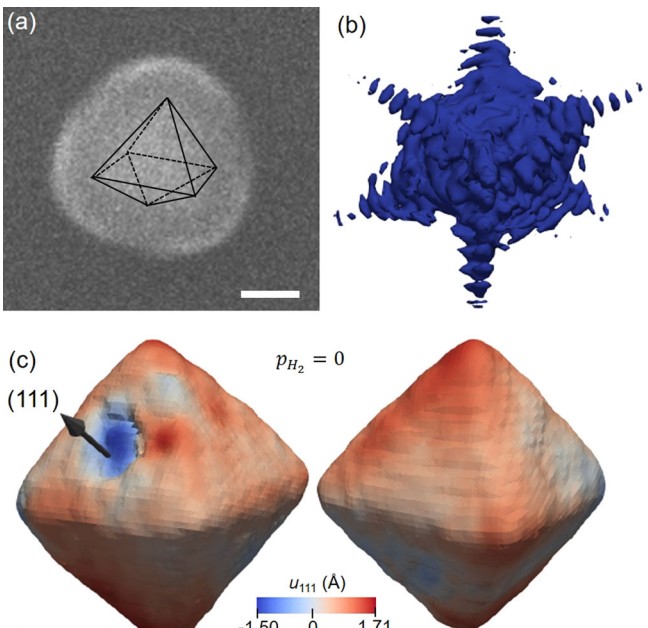

**Fig. 1 Imaging the displacement field of the octahedral Pd nanoparticle. a** Scanning electron microscopy image of an individual Pd nanoparticle, showing the $TiO_2$ layer surrounding the metallic core. The black lines are a guide for the eyes, distinguishing the Pd surface and the titanium dioxide. **b** Isosurface of the 3D diffraction pattern obtained from the Pd nanoparticle measured under the pristine state ($p_{H_2} = 0$). **c** Different views of the reconstructed displacement field for the pristine state nanoparticle shown as an isosurface, colored according to its displacement component along the [111] direction, $u_{111}$. The black arrow represents the [111] direction of the measurement Q-vector. The scale bars in (**a**) and (**c**) represent 100 nm.

the [111] direction, and negative values (blue-colored), along the opposite direction. The displacement field varies significantly over the nanoparticle surface, indicating the presence of strain that may have been introduced on the nanocrystal during the synthesis and/or ALD procedures. These strong inhomogeneities can be clearly seen in the cross-section views of the lattice displacement maps shown in Supplementary Fig. 3. The crystal size is ~330 nm from the top to the bottom corner. The particle volume is $9.2 \times 10^6$ nm$^3$ estimated with *Paraview*[30] at an isosurface contour level of 20%. The reconstructed electron density for the nanocrystal measured under these conditions, i.e., RT under nitrogen gas, corresponds to the volume of the metallic phase of palladium. In the first view of the reconstruction shown in Fig. 1c, we observe a concave depression of the surface present on the facet containing the (111) Q vector. This feature and the strong phase reversal associated with it were seen in the images using a wide range of parameters in the reconstruction algorithm, meaning that it is not a reconstruction artifact, but a real feature. The presence of this concave region increases the surface to volume ratio, providing further active sites for chemical reactions[31].

**Locating the seed inside the Pd octahedral nanoparticle: experimental and FEA calculations results.** Figure 2a, b shows the 3D reconstruction plotted as a semitransparent isosurface for the nanoparticle in its pristine state. In Fig. 2a, the plotted displacement was thresholded for negative values ($< -0.36$ Å, blue-colored). It is noticeable that the portion of the nanocrystal within this negative range is present close to the concave portion, in the direction of the

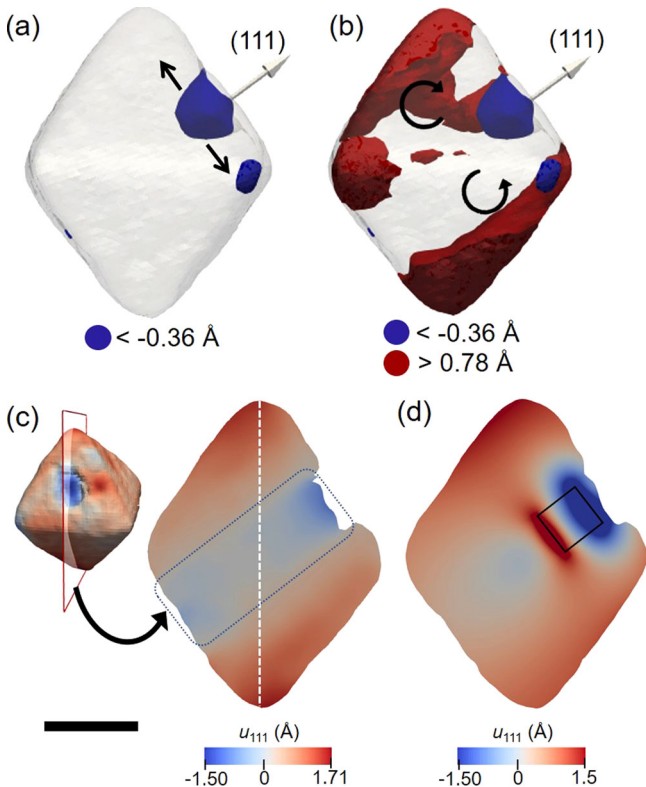

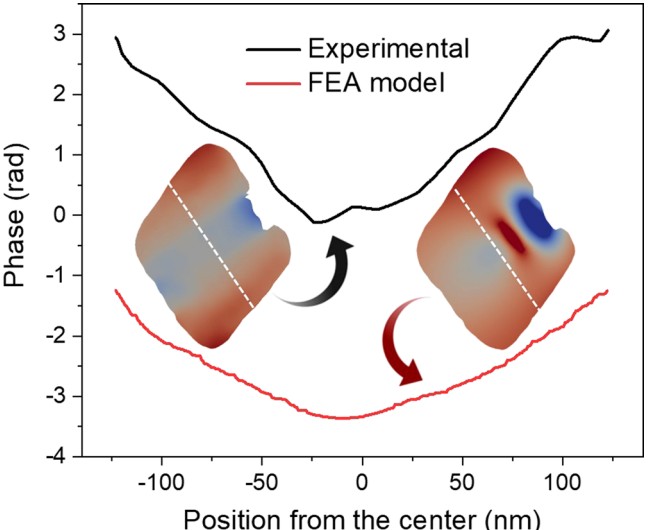

**Fig. 3 Phase line plot for the experimental data and the FEA model.** The insets are cross-sectional views of the experimental (middle-left) and FEA model (middle-right). The white dashed lines represent the position of the phase line plot for the experimental result (solid black line) and FEA model (solid red line).

**Fig. 2 Displacement field distribution over the Pd nanocrystal for the measurement done under pristine conditions. a** Semitransparent 3D plot of the Pd nanoparticle showing negative displacement values (<−0.36 Å, blue-colored) and (**b**) the same negative values shown in (**a**) and positive displacement values (>0.78 Å, red-colored). **c** Cross-section view of the internal displacement field distribution taken along the (111) Q vector. The position of the plane taken from the 3D reconstruction is shown in the top-left. **d** Cross-sectional view of the displacement distribution from the finite element analysis, modeled using the original reconstructed octahedral nanoparticle shape with a cube placed inside it to represent the seed nanoparticle. The square inside the plot represents the cube position. The scale bar in (**c**) denotes 100 nm and it is representative for (**a**)–(**d**).

Q vector and it is homogeneous, having its own crystalline identity. Because it has the expected size and roughly cubic shape, we identify this feature as the crystal-growth seed inside the larger nanocrystal. Interestingly, the Pd seed is not located in the center of the larger nanocrystal, suggesting a growth mechanism different from the conformal epitaxial growth, expected and often observed for seed-mediated growth. This off-center seeding behavior was already seen by Jin et al.[32] in the epitaxial growth of copper on Pd cores to produce core-shell nanocubes. They called this mechanism "localized epitaxial growth", and attributed it to the coalescence of different Cu shell domains, that nucleated and grew on different parts of the Pd seed. As a result, most of the Pd seeds were not located in the center of the final nanoparticles. Zeng and coauthors[33] also reported similar results concerning Pd@Ag core-shell nanoparticles. In this case, the asymmetric growth pattern was attributed to the moderate speed of the injection rate of the metal precursor solution, $AgNO_3$, into the as-synthesized Pd seeds.

In Fig. 2b, the displacement shown in Fig. 2a and positive displacement values (>0.78 Å, red-colored) are displayed. It is clear that the highly positive displacement values are concentrated close to the top and bottom portions of the nanocrystal, as best visualized in the cross-section shown in Fig. 2c, taken in the position of the transparent plane with red edges (top-left), along the Q vector direction. To explain this displacement pattern, we

suggest that the seed would apply an outward force on the nanocrystal, in the sideways direction, as indicated by the black arrows in Fig. 2a, and the nanocrystal would react by rotating its two sides, towards the direction of the Q vector, as shown by the black circular arrows in Fig. 2b. To go further in our investigation, we performed FEA calculations using the COMSOL multiphysics software package[34]. The model nanoparticle consisted of a cube (representing the seed) inserted inside the original octahedral nanocrystal shape, imported from its 3D reconstructed image. Both the crystal and seed were specified as metallic Pd, but different coefficients of thermal expansion (CTE) were assigned to them. Changing the temperature of the simulation allowed the differential distortion pattern to be tuned and visualized. A detailed description of the FEA modeling is presented in the "Methods". Figure 2d shows the displacement distribution of the x component (corresponding to the direction of the Q-vector in our experimental data) as a cross-sectional view of the 3D model Pd nanocrystal. The black line square represents the cube position in our model. It can be seen that positive displacement values are pronounced near the top and bottom edges of the nanocrystal, relative to the center, in agreement with the displacement distribution seen for the nanocrystal measured under pristine conditions. Figure 3 shows the phase line plot over the white dashed lines drawn in the nanocrystal's cross-sections (inset) of the experimental data (solid black line) and the FEA model (solid red line). A similar trend is seen in both experimental and theoretical data, with phase values more pronounced in the extremities of the crystal and smoother in the center region, with a distinct "V"-shaped displacement pattern.

Figure 4b, c shows cross-sectional views of the experimental data and FEA model, respectively, taken in the positions shown in Fig. 4a. The overall displacement field resemblance is remarkable, with a strong displacement pattern in the center, propagating in 3D through the direction of the seed in a cylindrical form. This pattern indicates that the seed acts exerting pressure in the nanocrystal in the outwards direction, leading to the displacement pattern seen. The red region seen in the center of the cross-section number 2 shown in Fig. 4c, disagreeing with the blue result of the experiment, is attributed to limitations of using a negative CTE to model the seed as a cube in the FEA. In this case,

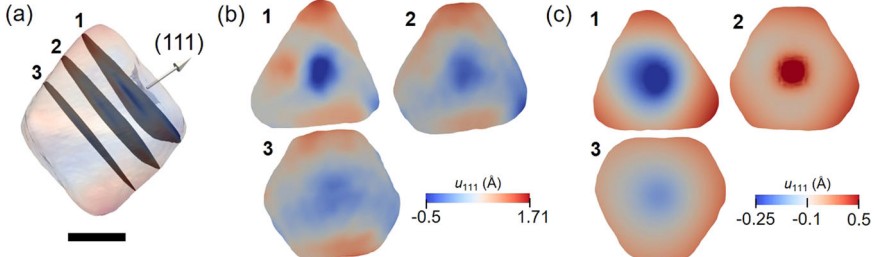

**Fig. 4 Cross-sectional views of the displacement distribution for the experimental data and the FEA model. a** Spatial position of the cross-sections (1–3) shown in (**b**) and (**c**). The nanocrystal is plotted as a semitransparent phase isosurface and the arrow represents the (111) scattering vector. **b** Cross-sectional views for the experimental data and (**c**) the same slices for the FEA model. The scale bar in (**a**) denotes 100 nm.

the cross-section position was taken at the end of the cube, and as the cube shrinks during the modeling process, this portion of the crystal will move forward in the positive direction of the $x$ component. The FEA modeling was also done using an ideal symmetric octahedral shape (Supplementary Fig. 4), using the same approach: a cube representing the seed was introduced inside the octahedron and both were denominated as Pd. After applying a different CTE for both at a given temperature, the displacement field pattern is very similar to the experimental results. The agreement between the experimental result and FEA calculations validates the capability of BCDI to give displacement distribution with subangstrom sensitivity.

For bimetallic nanoparticles in a core-shell arrangement, a substantial lattice mismatch between the two metals is expected to induce significant strain and possible defects in the crystalline structure over several atomic layers around the interface. In the case of the seed-mediated approach used in this work, the seed nanoparticle and the larger nanocrystal are the same material, so perfect epitaxial growth is expected. Even though the lattice-match criterion is met, issues related to the presence of an interfacial layer can interfere with perfect epitaxial growth on the seed surface. In the present work, the seed precursor solution was stabilized by $CTA^+Br^-$ bilayers on the surface of Pd cubes. Iodide ions were then added (see the details in the "Methods") and, due to the higher adsorption strength of iodide ions than that of bromide[35], a $CTA^+I^-$ bilayer is expected to be formed on the seed surface. When this solution was directly transferred to the so-called growth solution, the presence of the bilayer can influence the nanocrystal growth from the seed surface. In this BCDI work, we demonstrated that the resulting larger crystal is highly distorted by the presence of the seed. To the extent of our knowledge, this is the first work reporting a three-dimensional image of a mono-metallic seeded nanocrystal, where the seed location can be identified inside the nanoparticle.

**Imaging the electronic density and mapping the displacement field in 3D during the hydride phase transformation**. After the pristine state measurement, the pressure inside the cell was increased in one step using a gas mixture of 4% of hydrogen in nitrogen, corresponding to an approximate hydrogen partial pressure of 4410 Pa. Then, the same nanoparticle was measured at RT over time. The reconstructions displayed in Fig. 5 show the evolution of the nanoparticle morphology, after hydrogen exposure. The images in states *I*, *II*, and *III* were generated by averaging multiple diffraction scans over different hydrogen exposure times: from 4 to 16 min, from 19 to 28 min, and from 31 to 40 min, respectively. The first pattern taken after 1 min of hydrogen exposure was discarded as the pressure inside the cell was probably not stable. In fact, the integrated intensity for this scan is much weaker if compared with the following ones (scans from

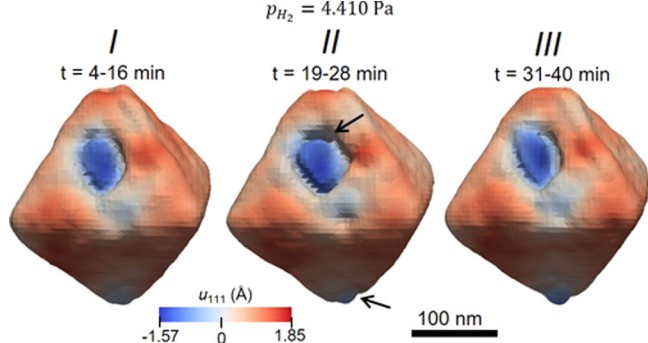

**Fig. 5 Displacement field evolution during the hydrogen exposure measurement ($p_{H_2}$ = 4.410 Pa).** Time evolution of the morphology and surface displacement distribution for the same Pd nanoparticle measured under a hydrogen gas atmosphere. The hydrogen exposure time ($t$) is indicated.

time = 4–40 min, displayed in Fig. 5), as shown in Supplementary Fig. 5.

The displacement values shown in Fig. 5 were unwrapped, and the original wrapped phase for states *I–III* are shown in Supplementary Fig. 6. In Bragg geometry, the electron density derived from the reconstruction comes from the atomic planes in that particular Bragg orientation. The missing Bragg electron density regions in the reconstructions suggest a transition from the Pd hydrogen-poor α phase to the hydrogen-rich lattice expanded β phase, the latter being formed for compositions $PdH_x$, $x > 0.017$[36]. While the lattice constant for palladium in the α phase is 3.89 Å, very close to that of the metallic state, it is 4.03 Å for the β phase which has a different cubic crystal structure with a large number of ordered hydrogen atoms[28]. Therefore, the two phases will appear in different positions in reciprocal space. The (111) Bragg peak of the α phase was centered on the detector, but the one corresponding to the β phase was not seen because it was outside the area detector's angular range. Thus, the disappearance of the electron density, mostly in the top tip region in Fig. 5, starting in state *I*, is interpreted as due to the formation of the β phase in this region. Smaller gaps due to the hydrogen-rich β phase are also seen in the concave region in the center of the (111) facet containing the Q vector and also in the bottom tip of the octahedron, as indicated by the black arrows in Fig. 5. Despite the transformation from α to the β phase is very likely to occur, the detector did not extend far enough to record the diffraction pattern of the hydrogen-rich phase.

For states, *I* and *II* the volume percentage of the β phase is 2.5%, and for state *III*, it is 4.5%, predominantly allocated at the top corner of the octahedron. This indicates that the hydrogen is concentrated in the tip regions where there is a high number of

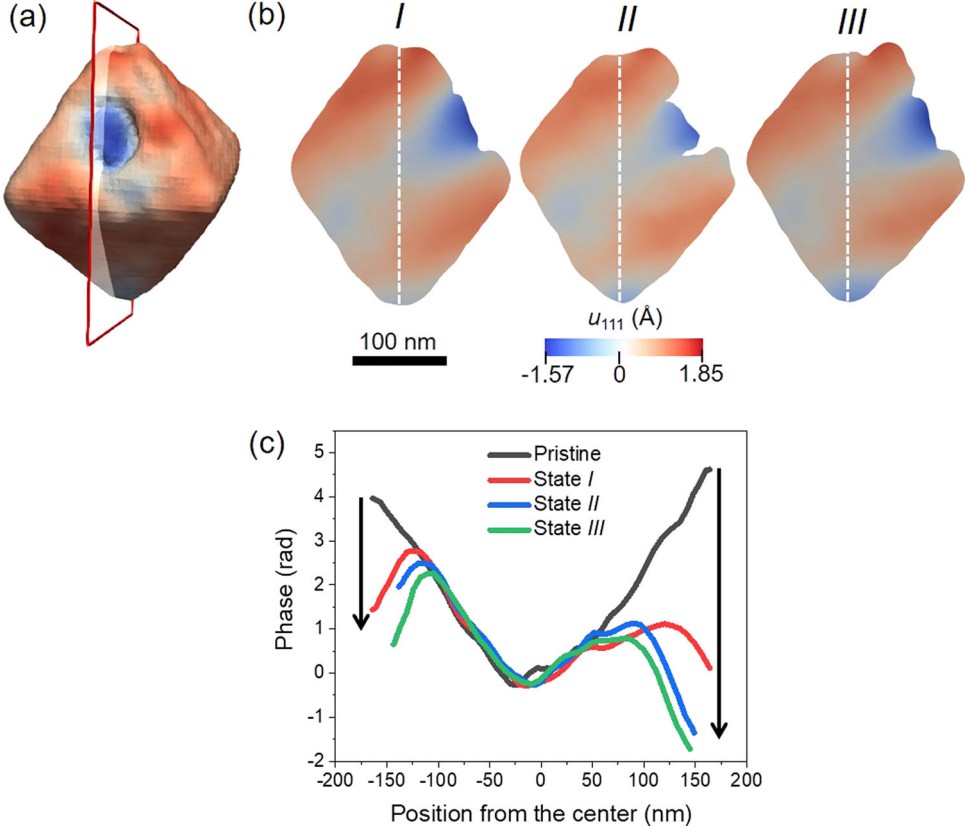

**Fig. 6 Cross-sectional views of the phase distribution for states I–III. a** 3D reconstruction with the transparent plane representing the position of the cross-sections shown in (**b**) for states I–III. **c** Phase plotline over the dashed white line shown in (**b**) for states I–III. The black arrows at the sides of the plot show the direction of the phase reversal.

lower coordinated Pd atoms. This result is in accordance with the "cap model"[26], which proposed a two-phase-morphology, with the second phase capping the first and a planar interface between the two phases. The authors theoretically calculated the elastic energy for a 10% volume fraction of β phase distributed both as a spherical-cap and as a core-shell structure. For all hydrogen concentrations, the cap model led to smaller strain energy. This model is in conformity with the results seen here, where the β phase nucleates at one corner of the octahedral nanoparticle. Similar findings were reported for the hydrogenation of octahedral and cubic-shaped Pd nanocrystals[7,25,28]. In a series of systematic scanning transmission electron microscopy and electron energy loss spectroscopy measurements, Sytwu et al.[28] found similar results for cubic- and octahedral-shaped Pd nanocrystals, with an α–β phase interface moving linearly in time, indicating that the reaction is rate-limited by some surface reaction mechanism and not by simple diffusion. The corner-initiated nucleation is most likely due to either the high number of lower coordinated Pd atoms in these regions or tensile strain at the corners, resulting from the α phase formation in these areas[28]. In fact, the presence of strain can affect hydrogen diffusivity. For example, Kimizuka et al.[37], with an ab initio molecular dynamics model, found that for H atoms in Pd, a hydrostatic strain of 2.4% enhanced the hydrogen lattice diffusion rate by around 20 times that of unstrained Pd at RT. In the β phase of palladium hydride, the H atoms occupy two possible interstitial sites in the fcc lattice structure of Pd: octahedral (O) and tetrahedral (T). The diffusion process at the atomic level occurs by thermally activated jumps of H atoms between T and O sites, being the O-sites more thermodynamically stable[37–39] and therefore, for the bulk β phase, only the O sites are occupied by the H atoms under ambient conditions[39,40]. The difference between the energy for hydrogen in O and T sites is substantially decreased as the strain increased in the Pd crystalline structure[37]. Consequently, besides the O sites, the T sites can be occupied as well, the latter being even more stable than the O sites for strain values >1.2%. Quasielastic neutron scattering experiments[39] revealed that for Pd nanocrystals of 8 nm diameter, 30% of the H atoms are located at the T sites. Because an increase in the Pd lattice constant was seen in the pristine Pd nanoparticles[39], lattice distortions are expected to play an important role in the dynamics of the diffusibility of hydrogen.

Figure 6a shows the 3D reconstruction for state I, where the transparent plane with red edges is displayed as cross-sections in Fig. 6b for states I, II, and III. All the cross-sections were taken at the same position. The displacement field distributions are significantly modified from the pristine state measurement (Fig. 2c), mostly in the vicinity of the top and bottom nanocrystal tips. This is very clear in the phase plot line shown in Fig. 6c, which represents the phase distribution along the white dashed lines drawn in Fig. 6b. A remarkable feature of this plotline is that the phase values at the edges of the particle tend to reverse direction progressively with hydrogen exposure, as indicated by the black arrows in the plot. There is a clear ramping behavior in the phase with abrupt reversals. In BCDI, phase ramps are a direct indication of regions of a crystal with a slightly different lattice parameter, so we can describe the overall distribution seen as blocks of α phase going forward from the top and bottom terminations of the nanocrystal, while the rest of the nanoparticle remains as metallic Pd. This result is in agreement with the preferential hydrogen uptake from the corners of the octahedron, as discussed above.

In this study, we have measured in situ the 3D displacement field distribution in a single octahedral Pd nanoparticle under pristine conditions and during the α–β phase transformation induced by hydrogen exposure. The reconstruction revealed the presence of a concave portion on one of the 111 facets, and the region right below it presents a homogeneous displacement distribution that is identified as the seed precursor used in our nanoparticle synthesis. We found that the presence of the seed has an influence on the 3D displacement distribution throughout the volume of the pristine nanocrystal. This characteristic displacement field was fully explained by finite element calculations, modeled with a cube inserted into an octahedral Pd nanoparticle causing rotations of the outer regions. During the hydriding phase transformation, we were able to capture the reaction over time. The reaction stages (states I–III) demonstrate that the hydrogen-rich β phase is predominantly formed on one tip of the nanoparticle, where there is a high number of undercoordinated Pd atoms, and pronounced displaced atoms. Here we study the dynamics of the hydrogen poor-hydrogen rich phase transformation and provide important insights into the investigation of dynamic processes in individual nanocrystals under realistic conditions. Hydrogen solubility and diffusivity are properties determining how much and how fast the hydrogen will be delivered in the structure and, in this work, we have shown that both properties are highly influenced by the lower coordination number in the vertices rather than surface faceting.

## Methods

**Palladium nanocrystals synthesis**. The octahedral palladium nanoparticles were synthesized following a seed-mediated growth approach based on a previous report[41]. Cube-shaped palladium seed nanoparticles were synthesized by mixing 0.500 g of CTAB, 0.5 mL of a 0.01 M tetrachloropalladinic acid ($H_2PdCl_4$) solution, 0.3 mL of a 0.01 M sodium iodide (NaI) solution, and 10 mL of nanopure water. The resulting solution was maintained in an oil bath for 5 min at 95 °C. Then, 200 μL of a 0.04 M ascorbic acid solution was added and the temperature was kept at 95 °C for 1 h. The growth procedure was done by adding together 0.360 g of CTAB, 0.250 mL of a 0.01 M $H_2PdCl_4$ solution, 0.050 mL of a 0.001 M NaI solution, and 9.375 mL of nanopure water. This solution was maintained at 30 °C. After 5 min, 80 μL of the as-prepared Pd seeds and 250 μL of a 0.04 M ascorbic acid solution were added and the system was maintained at 30 °C for 40 h.

**TiO₂ atomic layer deposition**. The crystalline $TiO_2$ coating was deposited in a Cambridge nanotech (Savannah 100) system based on a previous report[42]. The reaction was performed at 275 °C with a constant flow of $N_2$ at 20 sccm (base pressure ~100 mTorr). Titanium isopropoxide (Ti(iPrO)$_4$) (heated to 75 °C) served as the Ti precursor and nanopure $H_2O$ at RT was used as the oxygen precursor. The pulse and purge times for Ti(iPrO)$_4$ and $H_2O$ were 0.1 s and 5 s and 0.01 s and 10 s, respectively.

**TEM and SEM imaging**. For the electron microscopy imaging of the seed nanoparticles, 5 μL of a concentrated seed solution was drop-casted onto carbon-coated copper grids and left undisturbed for drying at RT. TEM bright-field images were acquired by a JEOL JEM-2100F microscope operating at 200 kV. The $TiO_2$ coated Pd nanoparticles were imaged using a Hitachi FlexSEM 1000, operating at an accelerating voltage of 15 kV. In this case, the sample was prepared by drop-casting 10 μL of the colloidal dispersion on Si wafer, which was left to dry at RT prior to analysis.

**BCDI experiment details**. The experiment was performed at 34-ID-C at APS, Argonne National Laboratory, USA. A double crystal monochromator selected the energy to 9 keV and Kirkpatrick–Baez mirrors were used to focus the beam to $600 \times 600$ nm$^2$. The coherent diffraction patterns for the Pd (111) Bragg reflection was recorded in a Medipix detector ($512 \times 512$ pixels, $55 \times 55$ μm$^2$), which was placed 0.5 m away from the sample. The nanocrystals were drop-casted on a silicon wafer substrate, which was placed inside an in situ cell. The rocking curve of the Pd Bragg peak was collected by using angular steps of 0.02° in 60 frames, and 2 s of exposure time. The patterns were recorded continuously during the hydrogen exposure. The Pd α–β phase transformation was induced by flowing a gas mixture composed of 4% of hydrogen gas in $N_2$ in an approximate partial pressure of $H_2$ of 4410 Pa.

**Data reconstruction parameters**. Error reduction (ER) and hybrid-input-output (HIO) algorithms were used for the reconstructions, in a total of 620 iterations[13,43]. The phase retrieval code was initiated with 20 ER iterations, alternating with 180 iterations of the HIO algorithm, starting from random phases. A gaussian shrink-wrap function with a threshold of 0.15 was used as the support constraint[44]. The reconstructions were plotted as isosurface contour maps of the image amplitude in 3D and 2D cross-sections with *Paraview*[30]. The maps were colored with the image phase, which represents the projection of the local displacement onto the Q-vector.

**3D phase unwrapping method**. The 3D phase unwrapping algorithm is performed based on a non-continuous path to complete the unwrapping process[45]. The algorithm is used to remove the $2\pi$ phase discontinuity of the phase extracted from complex volumetric data. The algorithm firstly calculates the reliability of each voxel using the second difference method. Then, with the obtained horizontal, vertical, and normal edges' reliability, the algorithm will sort all of these edges according to their reliabilities. Finally, the algorithm will unwrap voxels according to the edges reliabilities by offsetting these voxels with a multiple of $2\pi$ to make the input phase change continually.

**Theoretical approach modeling**. FEA was done using COMSOL Multiphysics software[34]. The FEA simulations accounted for materials properties including domain temperature, thermal expansion, and three-dimensional object shape. A cube with 50 nm edge dimensions was united together with the octahedral original shape, imported from the BCDI reconstruction, and both were denominated as metallic Pd. In order to simulate the displacement field distribution caused by the presence of the seed, the cube was given a different anisotropic coefficient of thermal expansion (CTE), set to $\alpha_{xx} = 0$, $\alpha_{yy} = \alpha_{zz} = -4 \times 10^{-4}\,K^{-1}$ instead of the original CTE for metallic Pd, which is $1.18 \times 10^{-5}\,K^{-1}$. The CTE value was set to 0 for the octahedral domain, and the temperature was raised by 100 K in both domains. Thermal expansion, which is feasible to model by FEA, is used as a surrogate for interfacial strain. We do not know the details of the structure of the interface between the seed and the octahedral-grown crystal, which could include lattice misfits, trapped solvent molecules, or ligands associated with the growth. All of these contribute to strain. We introduced strain by using differential stress with a null CTE on one side of the interface and a negative CTE on the other. A second FEA approach was to build an ideal symmetric octahedron plus the cubic shaped object in AutoCAD[46], that was load in COMSOL Multiphysics, and then the model was built similarly as the one using the original data, except that the CTE ($-1 \times 10^{-4}\,K^{-1}$ in this case), was defined isotropically.

## Data availability

The data discussed in this paper are available upon request to the corresponding authors. Correspondence and requests for materials should be addressed to A.F.S. (e-mail asuzana@bnl.gov) and I.K.R. (e-mail: irobinson@bnl.gov).

## Code availability

The code used for the 3D phase unwrapping is available upon request to L.W. (lwu@bnl.gov).

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

## Acknowledgements

A.F.S. and I.K.R. would like to thank Xiaojing Huang for the help with the electron microscopy. B.P.W. and C.-K.T. would like to thank the support from Boston College. Thank you to Connor C. Starkey for assistance with the Pd nanoparticle synthesis and to Haochuan Zhang and Dunwei Wang for assistance with ALD coating. This research used resources of Brookhaven National Laboratory, supported by the U.S. Department of Energy (DOE), Office of Basic Energy Sciences, Division of Materials Sciences and Engineering, under Contract No. DE-SC0012704. The BDCI experiment took place at the Advanced Photon Source, a U.S. DOE Office of Science User Facility operated for the DOE Office of Science by Argonne National Laboratory under Contract No. DE-AC02-06CH11357.

## Author contributions

A.F.S., L.W., T.A.A., R.H., W.C., and I.K.R. performed the experiment. C.-K.T. and B.P. W. synthesized the sample. C.-H.K. performed electron microscopy imaging. L.W. wrote the code for the phase unwrapping procedure. A.F.S. and I.K.R. conducted the finite-element analysis modeling. All the authors interpreted the results. A.F.S. and I.K.R. wrote the paper and all authors contributed to editing the manuscript.

## Competing interests

The authors declare no competing interests.
