## [Peer Review File · Communications Chemistry]

Reviewers' comments:

Reviewer #1 (Remarks to the Author):

Report on "Structure of a single palladium nanoparticle and its dynamics during the hydride phase transformation" presented by A. F. Suzana et al., for publication in Communications Chemistry

I have read with a lot of intention the paper presented by A. F. Suzana et al. This manuscript describes a structural analysis of a palladium nanoparticle, produced with a seed-mediated approach and further exposed to hydrogen environment. The study is based on x-ray coherent diffraction imaging able at producing images of the strain distribution within the particle, in situ, in 3D. The experimentally obtained images are compared to FEA calculations. The main results of the article are (i) the observation of the seed mediation of strain in the pristine case and (ii) the observation of the formation of the hydride phase occurring at specific regions of the particle. The manuscript is extremely well written, presents sound results and is based on a challenging experimental approach. There is now doubt that it could be published with some revisions detailed further down in this report. However I am wondering whether it reaches the standard for publication in Communications Chemistry, for the three main reasons detailed hereafter:

1 - Lack of novelty: As mentioned by the authors, a similar study (same scientific question, same experimental approach) was performed a few years ago by Ulvestadt et al. (see e. g., Ref. [23, 24, 25], in the manuscript). In their present work, the authors underline the fact that the nanoparticles have been obtained by seed-mediation, which clearly differs from the preparation process used to produce the nanoparticles used by Ulvestadt et al. Here, they further observe specific strain field associated with the seed. The novelty being here, it would have been interesting to perform a thorough study of the seed effect to study its impact on e.g., the final particle size, etc.

2 - Lack of relevance: this work aim at providing new insights on a systems which is known to present deviation due to size effects with respect to its bulk counterpart. However, the size-induced effects are expected to occur below 110 nm (from [4]), or 80 nm (from [6]) or 30 nm (from [9]). The particle analyzed in this work is about 330 nm large. Then, I could not convince myself of the relevance of the study with respect to the arguments presented from line 46 to line 60.

3 - Lack of statistical robustness: interesting results have been produced on the formation of the hydride phase, in this work. In their experimental approach, the authors use a time-averaged data set calculated from a series of intensity acquired as a function of time. They define three time windows: how were chosen the time windows? How would they vary as a function of the particle size, the seed, the hydrogen pressure, etc.? How do the findings observed on a single nanoparticles help to progress on the general knowledge in the related field?

Beside these three main points, I also have a few major and minor questions and comments detailed below:

A - Major comments and questions

#1 - Lines 46-60: Specify the different sizes related to the cited references [4]-[9].

#2 - Line 60: "... picometer sensitivity" instead of "resolution".

#3 - Lines 125-127: The effect of TiO₂ capping onto the diffusion process is assumed to be negligible: what evidence could the authors provide to support this assertion?

#4 - Lines 129-139: Please specify the sizes of the produced particles.

#5 – Line 193: Could the authors justify their choice regarding the CTE values in the seed and surrounding crystals, except on the fact that it better reproduces the data? What could be responsible for different CTEs in these two Pd structures?

#6 – Lines 240 – 246: I am not convinced by the averaging of the intensity patterns. Indeed, the intensity is the square of the diffracted field, which is itself the Fourier transform of the sample exit field. Hence, the intensity and the sample exit field are not linked by a linear relationship. When the average of intensity is calculated, it does not correspond to an averaged sample, in particular if the intensity is (slowly) changing. This leads to one of my main comments detailed above. How are the time windows defined? How do the different intensity patterns evolve with time (center of mass of Bragg peak? width of Bragg peak? extend of the streaks? Etc.). Is it possible to produce a time-dependent figure, based on the raw data and helping to understand why the time windows were defined as proposed?

Furthermore, why not inverting all diffraction patterns independently and calculating the average of the particles, using the successively retrieved particle images? A routine to check the likelihood between two successively retrieved particles could be introduced to better justify the chosen time windows.

#7 – Line 311-312: ‘... our result show an unambiguous methodology’: The methodology, which is not new, has not been demonstrated with this work. It is suggested to tone this statement down.

B – Minor comments and questions

#1 – Lines 46 – 48: A reference is needed here.

#2 – Lines 73 -78: Please, introduce the intensity pattern, to better highlight the problem of the missing phase.

#3 – Line 77: Please, introduce the constraints in real and reciprocal spaces, otherwise the reason why it converges (and not stagnates) is difficult to understand.

#4 – Lines 80-81: Please, specify that the retrieved quantity is the projection of the displacement field onto the Bragg vector (and not only the displacement field).

#5 – Line 81: Instead of ‘equilibrium position of the atoms’, rather write the ‘an ideally homogeneously strained crystal’ or ‘an ideal homogeneous crystal’.

#6 – Line 246: While the transition from the alpha to the beta phase is the most likely, it is not demonstrated here (using e.g. a diffraction pattern measured at the end of the experiment, showing the presence of a beta diffraction peak). It is suggested to be more cautious here and mention other possible effects to explain the missing electron density...

Reviewer #2 (Remarks to the Author):

This manuscript reports an in situ Bragg CDI for the 3D displacement field distribution in a single octahedral Pd nanoparticle under pristine conditions and during the α - β phase transformation induced by hydrogen exposure. They were able to locate the seed nanoparticle in the final nanocrystal and to verify the origin of the strain distribution due to the seed in the larger nanocrystal by comparing the finite element model. They also observed that the hydride phase is predominantly formed at the low-coordinate sites of the nanoparticle.

Despite the fact that the results are impressive to see the seed in the final nanocrystal even with TiO₂ capping layer, the following issues need to be addressed.

- (1) It is unclear the justification of the FEA model with different anisotropic thermal expansion coefficients for the cube (i.e., the seed). In particular, without having the negative thermal expansion coefficient, such strain distribution cannot be obtained.
- (2) The main finding is due to a nanoparticle synthesized by seed-mediated growth. This should be reflected in the title.
- (3) The reference numbers should be checked. E.g., p. 3, lines 87 and 93, the Reference number should be 23 not 22.

Reviewer #3 (Remarks to the Author):

This paper reports the Bragg coherent diffraction imaging of the alpha-beta hydride phase transformation of palladium nanoparticle. The morphology and 3D displacement distribution of the nanoparticle under the phase transformation was visualized. Analysis was done in detail with the assistance of finite element calculations.

It was found that the volume change upon the transformation to the beta phase is predominantly allocated at the top corner of the octahedron, indicating that the hydrogen is concentrated in the tip regions. This is where lower coordinated Pd atoms exist, which is in accordance with the cap model predicted earlier [24].

They conclude that the diffusivity of the hydrogen atoms into the Pd crystalline structure is coupled with the coordination number and, possibly, lattice displacement. This seems to be consistent with previous works on the hydrogen diffusion in Pd crystals and Pd nanoparticles studied by first principle simulations [35] and neutron diffraction experiments [36-38].

The paper is convincing and interesting to the broad audience working on hydrogen in metals. Therefore, I think this paper could be published as is.

Dear Reviewers,

We would like to thank you for the valuable suggestions and comments. Please find below our reply to the reviewers' comments. We also provide a manuscript version with all the corrections marked in red, and one without the marking. We thank you for considering our revision for further evaluation.

On behalf of the co-authors,

Ana F. Suzana

Reviewers' comments:

Reviewer #1 (Remarks to the Author):

Report on "Structure of a single palladium nanoparticle and its dynamics during the hydride phase transformation" presented by A. F. Suzana et al., for publication in Communications Chemistry

I have read with a lot of intention the paper presented by A. F. Suzana et al. This manuscript describes a structural analysis of a palladium nanoparticle, produced with a seed-mediated approach and further exposed to hydrogen environment. The study is based on x-ray coherent diffraction imaging able at producing images of the strain distribution within the particle, in situ, in 3D. The experimentally obtained images are compared to FEA calculations. The main results of the article are (i) the observation of the seed mediation of strain in the pristine case and (ii) the observation of the formation of the hydride phase occurring at specific regions of the particle. The manuscript is extremely well written, presents sound results and is based on a challenging experimental approach. There is now doubt that it could be published with some revisions detailed further down in this report. However I am wondering whether it reaches the standard for publication in Communications Chemistry, for the three main reasons detailed hereafter:

1 - Lack of novelty: As mentioned by the authors, a similar study (same scientific question, same experimental approach) was performed a few years ago by Ulvestadt et al. (see e. g., Ref. [23, 24, 25], in the manuscript). In their present work, the authors underline the fact that the nanoparticles have been obtained by seed-mediation, which clearly differs from the preparation process used to produce the nanoparticles used by Ulvestadt et al. Here, they further observe specific strain field associated with the seed. The novelty being here, it would have been interesting to perform a thorough study of the seed effect to study its impact on e.g., the final particle size, etc.

Author reply:

Thank you for this observation. Indeed Ulvestad et al. have performed similar experiments on different-shaped crystals and published their work a few years ago. The investigation reported in our manuscript differs from the ones reported by Ulvestad et al. in two ways:

1. As the reviewer pointed out, we have an explanation for the 3D strain field, and it is associated with the seed nanoparticle inside the larger particle. Regarding the reviewer's comment on the study of the seed effect on the final particle size, etc., there are several works reporting the seed effect on the final particle. The main consideration to be highlighted is that the concentration of both the seed and the Pd metal salt added during the growth step play an important role in the final nanoparticle size. Regarding the shape of the final nanoparticles, the concentration of the NaI (sodium iodide) and the temperature are important parameters directly affecting the shape. Ref. [1] points to the effects of these varying parameters in the final nanoparticle; in this paper, the authors developed a versatile seed-mediated method to produce Pd nanoparticles with different shapes, simply by varying the concentration of KI (potassium iodide) and the temperature. The formation of different palladium facets was seen, depending on these parameters;

2. The particle shape investigated in our work is octahedral, which allows us to address the question of where the hydrogen absorption will take place in the crystal. In the as-mentioned work

by Ulvestad et al., the only anisotropies studied were due to the cubic shape [2]. The strains associated with the exposed vertices of the two crystal morphologies are strategically interesting and likely to respond differently: three facets meet at the vertex of a cube, while four meet on an octahedron.

[1] Niu, W. X.; Zhang, L.; Xu, G. B. Shape-controlled synthesis of single-crystalline palladium nanocrystals. *ACS Nano* 4 (2010).

[2] Ulvestad, A. et al. Avalanching strain dynamics during the hydriding phase transformation in individual palladium nanoparticles. *Nat. Commun.* 6, (2015).

2 – Lack of relevance: this work aim at providing new insights on a systems which is known to present deviation due to size effects with respect to its bulk counterpart. However, the size-induced effects are expected to occur below 110 nm (from [4]), or 80 nm (from [6]) or 30 nm (from [9]). The particle analyzed in this work is about 330 nm large. Then, I could not convince myself of the relevance of the study with respect to the arguments presented from line 46 to line 60.

Author reply:

We agree with the reviewer's comment on the fact that the size effects are expected for smaller particles. However, the six corners of the octahedral particle are regions with sharp features on the length scale of much smaller nanoparticles. They have a high number of lower coordination atoms and therefore these regions are expected to behave following similar size-induced effects to sub-100 nm particles. Indeed, the results seen in our work are a direct consequence of this: the hydrogen absorption and consequent formation of the hydride phase is very pronounced in the corners of the particle.

3 – Lack of statistical robustness: interesting results have been produced on the formation of the hydride phase, in this work. In their experimental approach, the authors use a time-averaged data set calculated from a series of intensity acquired as a function of time. They define three time windows: how were chosen the time windows? How would they vary as a function of the particle size, the seed, the hydrogen pressure, etc.? How do the findings observed on a single nanoparticles help to progress on the general knowledge in the related field?

Author reply:

We performed our BCDI measurement as multiple short scans of the diffraction pattern to have the possibility of seeing fast changes. When we did not see these, we averaged over different scans in order to improve statistically the results. This strategy was important in the case of our work, where the averaging was done as follows: after the hydrogen pressure was raised inside the in situ cell used in our experiment, we collect several scans over time. The first scan had to be discarded because this one represents the state for the first burst of hydrogen, when the pressure was not stabilized yet. The next scans (13 scans in total) were averaged in order to have the states equally spaced in time: 5 scans were averaged for state I (4-16 min), and 4 scans were averaged for each state II and III (19-28 min and 31-40 min, respectively). The results of Ulvestad et al. [2] are also presented in a similar way (averaging over the yellow points present in Fig. 1d in reference [2] above).

We believe that the time windows could be maintained for different particles (different sizes, shapes, etc.), using the same pressure. However, the response of the particle, i.e., the hydride phase composition, would vary depending on these changes. By changing the nanoparticle shape, and maintaining the time windows, we would expect different concentrations of α and β phases. Narayan et al., for instance, found that icosahedral Pd nanoparticles present reduced hydrogen storage capacity compared to cubes and prisms. By changing the particle size, we would expect a similar behavior, where smaller particles would present faster phase transformation, as already seen by Bardhan et al. [4] for Pd cubes. By changing the hydrogen pressure, we believe we could maintain the same time windows, but we would have more or less hydride phase depending on the pressure: a decrease in the hydrogen partial pressure would lead to either less hydride phase formation or no hydride phase. On the other hand, an increase in the hydrogen partial pressure would lead to more hydride phase.

Regarding the statistical robustness, we could only study one crystal at a time on each sample preparation, which changes irreversibly under hydrogen. We did have time to study one other crystal in a different experiment, and we could see a similar displacement distribution for the crystal measured under pristine conditions (partial pressure of hydrogen = 0), indicating the presence of the seed, as shown below:

We believe that the pronounced negative phase on one of the facets is the seed inside the nanocrystal.

[3] Narayan, T. C., Baldi, A., Koh, A. L., Sinclair, R. & Dionne, J. A. Reconstructing solute-induced phase transformations within individual nanocrystals. *Nat. Mater.* 15, 768–774 (2016).

[4] Bardhan, R. et al. Uncovering the intrinsic size dependence of hydriding phase transformations in nanocrystals. *Nat. Mater.* 12, 905–912 (2013).

Beside these three main points, I also have a few major and minor questions and comments detailed below:

A – Major comments and questions

#1 – Lines 46-60: Specify the different sizes related to the cited references [4]-[9].

Author reply:

The size of the nanoparticles of the different works were cited in the main text, as shown in the tracked text in red, lines 51-59.

#2 – Line 60: "... picometer sensitivity" instead of "resolution".

Author reply:

Thank you for this observation. The term was replaced in the text (line 62). The "resolution" of the technique usually refers to the spatial resolution, which is commonly in the 10-20 nm range.

#3 – Lines 125-127: The effect of TiO₂ capping onto the diffusion process is assumed to be negligible: what evidence could the authors provide to support this assertion?

Author reply:

We are assuming that the capping TiO₂ does not prevent the diffusion process since we can see lack of electron density in the reconstruction, a great indicative of the α - β phase transformation. Besides that, the small size of the hydrogen molecules makes them very likely to diffuse at first in the TiO₂ layer.

#4 – Lines 129-139: Please specify the sizes of the produced particles.

Author reply:

The size of the produced Pd particles (in the range of 200-300 nm) has been mentioned in the paragraph before (line 129 in the new file with the tracked changes).

#5 – Line 193: Could the authors justify their choice regarding the CTE values in the seed and surrounding crystals, except on the fact that it better reproduces the data? What could be responsible for different CTEs in these two Pd structures?

Author reply:

Thermal expansion, which is feasible to model by FEA, is used as a surrogate for interfacial strain. We do not know the details of the structure of the interface between the seed and the octahedral grown crystal, which could include lattice misfits, trapped solvent molecules or ligands associated with the growth. All of these contribute to strain. We introduced strain by using differential stress with a null coefficient of thermal expansion (CTE) on one side of the interface and a negative CTE on the other. We included this explanation on the current manuscript as well (lines 391-396).

#6 – Lines 240 – 246: I am not convinced by the averaging of the intensity patterns. Indeed, the intensity is the square of the diffracted field, which is itself the Fourier transform of the sample exit field. Hence, the intensity and the sample exit field are not linked by a linear relationship. When the average of intensity is calculated, it does not correspond to an averaged sample, in particular if the intensity is (slowly) changing. This leads to one of my main comments detailed above. How are the time windows defined? How do the different intensity patterns evolve with time (center of mass of Bragg peak? width of Bragg peak? extend of the streaks? Etc.). Is it possible to produce a time-dependent figure, based on the raw data and helping to understand why the time windows were defined as proposed? Furthermore, why not inverting all diffraction patterns independently

and calculating the average of the particles, using the successively retrieved particle images? A routine to check the likelihood between two successively retrieved particles could be introduced to better justify the chosen time windows.

Author reply:

The time windows were defined to improve statistics as explained above, they were chosen to be regularly spaced in time. The averaged intensity here regards the fact that a few scans were averaged for statistical improvement of the reconstruction, as highlighted in the reply above. The average over repeated scans also optimizes the signal-to-noise level of the coherent diffraction patterns. To correct for any sample drift during measurements, an automatic angular and position alignment step was carried out before every scan. In order to avoid phase ramps, re-centering the Fourier transform of the complex electron density took place during the algorithm reconstruction.

The routine used for the phase retrieval is a well established algorithm [5]. As the beam is not perfectly coherent, the algorithm also accounts for partial coherence effects, to accommodate both longitudinal and transverse partial coherence. The coherent diffraction pattern cannot be directly inverted, using an inverse Fourier transform, since the phase of the diffracted wavefield is not measured.

[5] Clark, J.N. et al., High-resolution three-dimensional partially coherent diffraction imaging, *Nat. Commun.*, 3 993 (2012).

#7 – Line 311-312: ‘... our result show an unambiguous methodology’: The methodology, which is not new, has not been demonstrated with this work. It is suggested to tone this statement down.

Author reply:

This sentence was rewritten: instead of “Our results show an unambiguous methodology to study the dynamics of phase transformations”, we rephrase it as “Here we study the dynamics of the hydrogen poor-hydrogen rich phase transformation... (lines 320 and 321 in the tracked file)”

B – Minor comments and questions

#1 – Lines 46 – 48: A reference is needed here.

Author reply:

A reference has been added to the sentence.

#2 – Lines 73 -78: Please, introduce the intensity pattern, to better highlight the problem of the missing phase.

Author reply:

Thank you for this observation. We added a sentence in this paragraph: “For nanosized objects, the coherent diffraction patterns contain fringes due to the finite size of the object, but in a diffraction experiment, only the amplitude of the complex Fourier is measured, not its phase.” (lines 77-79)

#3 – Line 77: Please, introduce the constraints in real and reciprocal spaces, otherwise the reason why it converges (and not stagnates) is difficult to understand.

Author reply:

The constraints were added in lines 82-85 in the new tracked file: “In this process, constraints in each space are adopted: the real space constraint includes a support region where the electron density of the object is allowed to exist, and the reciprocal space constraint consist of the solution having the observed amplitude of the oversampled diffraction pattern.”. A new reference for this statement was included as well.

#4 – Lines 80-81: Please, specify that the retrieved quantity is the projection of the displacement field onto the Bragg vector (and not only the displacement field).

Author reply:

We specified this in the corrected manuscript (line 87).

#5 – Line 81: Instead of ‘equilibrium position of the atoms’, rather write the ‘an ideally homogeneously strained crystal’ or ‘an ideal homogeneous crystal’.

Author reply:

The sentence has been rephrased: instead of “the phase is proportional to the displacement field of the crystalline structure compared with the equilibrium position of their atoms”, the new sentence is “the phase is the projection of the displacement field onto the Bragg vector of the crystalline structure compared with the ideal position of their atoms”.

#6 – Line 246: While the transition from the alpha to the beta phase is the most likely, it is not demonstrated here (using e.g. a diffraction pattern measured at the end of the experiment, showing the presence of a beta diffraction peak). It is suggested to be more cautious here and mention other possible effects to explain the missing electron density...

Author reply:

Despite the reconstruction clearly shows the presence of the missing Bragg electron density, indicating that this portion of the crystal has been transformed into the β phase, we modified the sentence in order to accommodate the reviewer’s concern. The new sentence is now: “Despite the transformation from α to the β phase is very likely to occur, the detector did not extend far enough to record the diffraction pattern of the hydrogen rich phase.” (lines 263-265)

Reviewer #2 (Remarks to the Author):

This manuscript reports an in situ Bragg CDI for the 3D displacement field distribution in a single octahedral Pd nanoparticle under pristine conditions and during the α - β phase transformation induced by hydrogen exposure. They were able to locate the seed nanoparticle in the final nanocrystal and to verify the origin of the strain distribution due to the seed in the larger nanocrystal by comparing the finite element model. They also observed that the hydride phase is predominantly formed at the low-coordinate sites of the nanoparticle. Despite the fact that the results are impressive to see the seed in the final nanocrystal even with TiO₂ capping layer, the following issues need to be addressed.

(1) It is unclear the justification of the FEA model with different anisotropic thermal expansion coefficients for the cube (i.e., the seed). In particular, without having the negative thermal expansion coefficient, such strain distribution cannot be obtained.

Author reply:

As explained above in the response to referee #1, we use thermal expansion, which is feasible to model by FEA, as a surrogate for interfacial strains that we cannot easily model. In order to accommodate the reviewer's concern, we also carried out FEA modeling using (a) isotropic CTE and (b) positive and isotropic CTE for both domains (seed and octahedral). The results are shown in the figure below and they match well the data. The anisotropic CTE in our model ($\alpha_{xx} = 0$, $\alpha_{yy} = \alpha_{zz} = -4 \times 10^{-4} \text{ K}^{-1}$) was adopted because the strain pattern in the x direction is very pronounced if α_{xx} is not set to 0 (see figure (a) below). We also modeled the system using positive CTE for both, the seed and the octahedra larger particle. The effect of this is to change the displacement direction in the final result (see figure (b) below). Therefore, the negative CTE is used in order to match the displacement direction seen in our experimental results.

(2) The main finding is due to a nanoparticle synthesized by seed-mediated growth. This should be reflected in the title.

Author reply:

We agree that the main finding of our work is due to the growth methodology employed. Therefore, we have modified the manuscript in order to reflect this already in the title.

(3) The reference numbers should be checked. E.g., p. 3, lines 87 and 93, the Reference number should be 23 not 22.

Author reply:

Thank you for this observation. All the references have been checked and corrected, including the one the reviewer points out.

Reviewer #3 (Remarks to the Author):

This paper reports the Bragg coherent diffraction imaging of the alpha-beta hydride phase transformation of palladium nanoparticle. The morphology and 3D displacement distribution of the nanoparticle under the phase transformation was visualized. Analysis was done in detail with the assistance of finite element calculations. It was found that the volume change upon the transformation to the beta phase is predominantly allocated at the top corner of the octahedron, indicating that the hydrogen is concentrated in the tip regions. This is where lower coordinated Pd atoms exist, which is in accordance with the cap model predicted earlier [24]. They conclude that the diffusivity of the hydrogen atoms into the Pd crystalline structure is coupled with the coordination number and, possibly, lattice displacement. This seems to be consistent with previous works on the hydrogen diffusion in Pd crystals and Pd nanoparticles studied by first principle simulations [35] and neutron diffraction experiments [36-38].

The paper is convincing and interesting to the broad audience working on hydrogen in metals. Therefore, I think this paper could be published as is.

Reviewers' comments:

Reviewer #1 (Remarks to the Author):

I thank the authors for their revision of the manuscript and their detailed explanations. Most of the questions and comments I had have been addressed carefully and in a convincing manner.

However, I am still concerned by the methodology, which consists in averaging the intensity patterns, instead of averaging the retrieved objects.

Indeed

$$\langle \text{object} \rangle_i = \langle \text{FT}(\text{diffracted_field}) \rangle_i$$

$$= \text{FT} \langle \text{diffracted_field} \rangle_i$$

$$= \text{FT} \langle \sqrt{\text{intensity}} \exp(i.\text{phase}) \rangle_i$$

with 'i' denoting the successive data acquisitions. This expression shows that $\langle \text{object} \rangle_i$ is not proportional to $\text{FT} \langle \text{intensity} \rangle_i$. Therefore the sum of the intensity patterns can not be used to retrieve an average object, unless the different intensity_i are almost not changing (so that one gets $\langle \text{intensity} \rangle_i \approx N \times \text{intensity}_1 \approx N \times \text{intensity}_N$).

In the previous report, I asked the authors if they could provide such evidence. At least they should show some time-dependent analysis of the raw data, to support or illustrate the validity of the assumption they make, when they average the intensity patterns over time.

A cleaner way to proceed would be to invert the intensity patterns successively and independently and to sum the retrieved objects to improve SNR.

Once this point is clarified, I believe the manuscript will be suitable for publication.

Reviewer #2 (Remarks to the Author):

The present version of the manuscript and the reply satisfy my comments and suggestions. Now I recommend that the paper can be published.

Reviewers' comments:

Reviewer #1 (Remarks to the Author):

I thank the authors for their revision of the manuscript and their detailed explanations. Most of the questions and comments I had have been addressed carefully and in a convincing manner. However, I am still concerned by the methodology, which consists in averaging the intensity patterns, instead of averaging the retrieved objects.

Indeed

$$\langle \text{object} \rangle_i = \langle \text{FT}(\text{diffracted_field}) \rangle_i$$

$$= \text{FT} \langle \text{diffracted_field} \rangle_i$$

$$= \text{FT} \langle \sqrt{\text{intensity}} \exp(i.\text{phase}) \rangle_i$$

with 'i' denoting the successive data acquisitions. This expression shows that $\langle \text{object} \rangle_i$ is not proportional to FT_i . Therefore the sum of the intensity patterns can not be used to retrieve an average object, unless the different intensity_i are almost not changing (so that one gets $\langle \text{object} \rangle_i \approx N \times \text{intensity}_1 \approx N \times \text{intensity}_N$). In the previous report, I asked the authors if they could provide such evidence. At least they should show some time-dependent analysis of the raw data, to support or illustrate the validity of the assumption they make, when they average the intensity patterns over time. A cleaner way to proceed would be to invert the intensity patterns successively and independently and to sum the retrieved objects to improve SNR.

Once this point is clarified, I believe the manuscript will be suitable for publication.

Author reply:

Thank you for the comment. We agree that the average diffraction of a strongly changing object will not produce a meaningful image. The following figure shows the plot Integrated intensity of the 3D data over time:

The first pattern (taken after 1 minute of hydrogen exposure) was discarded as it was the first diffraction scan recorded after the hydrogen exposure, and we were not sure if the hydrogen pressure was stable. In fact, the integrated intensity for this scan is much weaker if compared with the following ones (scans from time = 4-40 min). The following scans (representing States I, II and III) were averaged as explained in our previous reply letter. We can note some small differences in the integrated intensity of the diffraction patterns, but they are sufficiently consistent to allow averaging. When we examined images from inversions of individual frames, they were quite noisy and it was difficult to say if there were any changes between them. We looked at these first before deciding to group them to improve statistics.

Reviewer #2 (Remarks to the Author):

The present version of the manuscript and the reply satisfy my comments and suggestions. Now I recommend that the paper can be published.

Author reply:

Thank you for the comment and the positive feedback.

REVIEWERS' COMMENTS:

Reviewer #1 (Remarks to the Author):

I thank the authors for their honest answer and I suggest to better clarifying this question in the manuscript. Please add the plot you produced in your answer.

Reviewers' comments:

Reviewer #1 (Remarks to the Author):

I thank the authors for their honest answer and I suggest to better clarifying this question in the manuscript. Please add the plot you produced in your answer.

Author reply:

Thank you for your feedback. The figure has been added to the manuscript as the Supplementary Fig. 5, and we added a sentence in the manuscript file (lines 250-253) for the sake of clarification: "The first pattern taken after 1 minute of hydrogen exposure was discarded as the pressure inside the cell was probably not stable. In fact, the integrated intensity for this scan is much weaker if compared with the following ones (scans from time = 4-40 min, displayed in Fig. 5), as shown in Supplementary Fig. 5". All the changes we've done are marked in red in the manuscript and in the supplementary information file.